# Information and Self-Organization II: Steady State and Phase Transition

**DOI:** 10.3390/e23060707

**Published:** 2021-06-02

**Authors:** Hermann Haken, Juval Portugali

**Affiliations:** 1Center of Synergetics, Institute for Theoretical Physics, Stuttgart University, 70550 Stuttgart, Germany; hakenhermann@gmail.com; 2Department of Geography and the Human Environment, The Raymond and Beverly Sackler Faculty of Exact Sciences, School of Geosciences, Tel Aviv University, Tel Aviv 69978, Israel

**Keywords:** free energy principle, maximum entropy principle, Synergetics, what is life, pragmatic information

## Abstract

This paper starts from Schrödinger’s famous question “what is life” and elucidates answers that invoke, in particular, Friston’s free energy principle and its relation to the method of Bayesian inference and to Synergetics 2nd foundation that utilizes Jaynes’ maximum entropy principle. Our presentation reflects the shift from the emphasis on physical principles to principles of information theory and Synergetics. In view of the expected general audience of this issue, we have chosen a somewhat tutorial style that does not require special knowledge on physics but familiarizes the reader with concepts rooted in information theory and Synergetics.

## 1. Introduction

This is our second study on *information and self-organization*. In the first [1], our focus was on the exchange of information between a system and its environment. We did so from the perspective, firstly, of the Synergetics 2nd foundation (see definition below) and our *Information Adaptation* theory [2] with its notions of *Shannon information* (SHI), *semantic information* (SI), and *pragmatic information* (PI). Secondly, we did so from the perspective of SIRNIA that links the *synergetic inter-representation networks* (SIRN) approach with IA (information adaptation). We laid our emphasis on phase transitions, i.e., qualitative changes of the state of the system. In this second round on information and self-organization, we further elaborate on a complex system’s exchange of information with its environment, this time, however, in relation to other forms of exchange, namely of matter, energy, and entropy. We do so following Friston’s [3] *free-energy principle* (FEP), which entailed renewed interest in recent years in the notion of *free energy*, its relation to Schrödinger’s [4] question *What is Life* with his concepts of *negative entropy* as *free energy* (see below). Here, the focus is on the maintenance of a steady state.

This is also our second study that relates Friston’s FEP to Synergetic 2nd foundation and to our SIRNIA theory: Chapters 7, 8 of our book *Synergetic Cities* [5] explore the relationships, similarities and differences between Friston’s FEP, Bayes, Jaynes, Synergetic 2nd foundation, and SIRNIA. Thus, our discussion below starts with short reminders (Section 2): firstly, of Friston’s FEP, Synergetics 2nd foundation and SIRNIA—the basic theoretical frameworks at the center of this study, and, secondly, with the above noted comparison between them. Next, we look at Schrödinger’s [4] ‘what is life’ that, due to his emphasis on the properties of ‘openness’, negative entropy, free energy, and ‘order from disorder’, is considered one of the forerunners of complexity theory; and, we examine Friston’s attempt to respond to the question ‘what is life?’ from the perspective of his FEP. In Section 2.5, we point out that, because of insufficient information, complex systems can be dealt with only by a probabilistic approach. There are two pathways (micro/macro) to derive the appropriate probability distribution *P*, both of which will be followed up in the subsequent sections of our study. In Section 3, thus, we present the first pathway, namely the microscopic theory that derives P(q;Λ) of an open complex system by means of Synergetic’s first foundation. As is well recorded [6], Synergetics uses the phenomenon of the laser light as its canonical case study.

Central to Friston’s FEP is the notion of free energy. In Section 4, therefore, we elaborate on the notion of free energy and follow its metamorphosis from the original formulation of this concept by Helmholtz [7] in the context of Thermodynamics, to Feynman’s [8] formulation of it in the context of Statistical mechanics, to MacKay [9,10] (information theory), and, recently, to Friston in the context of life sciences. As it will transpire below, FEP as originally defined by Feynman, is a *terminus technicus* to characterize a specific mathematical procedure to calculate an energy, e.g., of a ferromagnet. In numerous other cases, including Friston’s, the quantity to be calculated is not an energy, but, e.g., numbers of populations of excited neurons, etc. Nevertheless, the formalism is that used by Feynman. We close Section 4 by a short discussion of the Synergetics 2nd Foundation, that is, Synergetics’ macroscopic approach that employs Jaynes’ Maximum (Information) Entropy Principle. 

In Section 5, we discuss another source of inspiration that enables to deal with complex systems—the Bayesian inference [11], that plays an important role in Friston’s FEP also. 

In Section 6, we return to the question of life that was mentioned above in connection with Schrödinger’s and Friston’s usage of the notion of free energy. This time, however, we look at the question of life from the perspective of the discussions in Section 2, Section 3, Section 4 and Section 5 above, focusing on the way a living system maintains its steady state. We first scrutinize the issue from Friston’s main example of the perception-action cycle. Secondly, we examine it from the perspective of Synergetics, and, thirdly, we compare the two approaches. Finally, we illustrate the latter by means of experimental results. 

We conclude our paper in Section 7 by emphasizing that homeostasis is not the only characteristics of life. Namely, to understand life, there is a need to deal with phase transitions, taking into account the fundamental role of information, and the principles of self-organization as theorized by Synergetics.

## 2. Reminders

### 2.1. A Concise Introduction of the Basic Terms

Friston’s FEP, Synergetic 2nd foundation, and SIRNIA are the basic theoretical frameworks at the center of this study, as noted. The FEP was proposed recently by Karl Friston as a unified brain theory “that accounts for action, perception and learning” [3] (Abstract). As such, it was applied to a variety of aspects in the domains of cognition, neurology, and biology (see Reference [12] for bibliography). Haken’s Synergetics [13], on the other hand, originated in physics but from the start was designed as a general theory of complex self-organizing systems. As such, it was applied to a variety of domains ranging from physics, through life sciences, sociology, psychology, cognition, AI among others, and also to cities [5]. This latter application to cities entailed the notions of SIRN, IA, and their conjunction SIRNIA. While there is no room in the present paper for a full-scale description of each, a few introductory words on these three basic terms is in place.

**Friston’s FEP.** The FEP is essentially a mathematical formulation of how complex, adaptive, self-organizing systems resist “a natural tendency to *disorder* … in the face of a constantly changing environment” [3] (p. 127). The principle that refers to biological systems, specifically to the dynamics of brains, suggests “that any selforganizing system that is at equilibrium with its environment must minimize its free energy …” (ibid). The principle further says “that biological agents must avoid *surprises* … [and, that] free energy is an upper bound on surprise, which means that if agents minimize free energy, they implicitly minimize surprise” (italics added). 

**Synergetics**, the “science of cooperation”, is Haken’s [6,13] theory of complex systems. It originated in physics with the aim to unearth the general principles underlying self-organization in open complex systems. Using the phenomenon of laser as its canonical case study, it describes the dynamics of such systems as a circularly causal process by which, firstly, the parts by means of their interaction give rise, in a bottom-up manner, to the global structure of a system, that can be mathematically described by an *order parameter* (OP). Secondly, once an OP emerges, it top-down prescribes the behavior of the parts—a process termed the *slaving principle*. This latter bottom-up “microscopic” approach is termed *Synergetics 1st foundation*. At a later stage, following the application of Synergetics to phenomena of cognition and brain functioning, the *Synergetics 2nd foundation* [14] was developed as a top-down “macroscopic” approach. Building on Shannon’s [15,16] theory of information, it theorizes about the way a complex cognitive system (e.g., a brain) extracts information out of the sparse or big data furnished by the environment and on the basic of this data behaves and acts. As can be seen, the two foundations complement each other: the 1st focuses mainly on the internal dynamics, while the 2nd adds the interaction with the environment. 

**SIRNIA**. The notions *SIRN*, *IA*, and their conjunction *SIRNIA* were designed to deal with a property that is specific to human agents as complex systems, namely that, as complex adaptive systems (CASs), they adapt not only by means of behavior and action, but also by the production of artifacts—the elements of culture and of cultural evolution. Thus, while all living systems (including humans) are subject to the slow process of Darwinian evolution, humans are subject also to the very fast process of cultural evolution—the evolution of artifacts [5,17,18]. The two components of SIRNIA were specifically designed to capture and model this process: SIRN by focusing on the exchange of information with the environment, while IA on the way the mind/brain/body process this information. Figure 1 describes this basic SIRNIA model.

In Figure 1, the outer solid line rectangle refers to SIRN, that is, to an agent who is subject to two flows of information: One that comes from the environment and one that comes from the agent’s mind/brain. The interaction between these two flows gives rise to behavior, action, and production in the environment, as well as to feedback information to the agent’s mind/brain, and so on, in circular causality. The inner dotted line rectangle refers to IA, that is, to that bottom-up process by which the agent’s mind/brain transforms the data flow from the environment into Shannonian information (SHI), which then triggers a top-down flow of semantic information (SI) that, by means of the inflation or deflation of the SHI, gives rise to two forms of output: SI that feeds back to the mind/brain, and pragmatic information (PI) in the form of behavior, action, and production in the environment, and so on, in circular causality. As can be seen in Figure 1, the SIRNIA model integrates the two foundations of Synergetics: the SIRN component is in line with the 1st, while the IA component with the 2nd. 

### 2.2. Previous Comparisons between Friston’s FEP and Synergetics’ 2nd Foundation

Complex systems are typically characterized by long periods of steady state interrupted by short periods of phase transition. From this perspective, Friston’s FEP refers, and adds insight, to the dynamics of steady states but not to that of phase transitions. It suggests that CASs (complex adaptive systems) have innate tendency to avoid phase transition and perpetuate their steady state. Synergetics accepts this view and indeed shows that steady states emerge and are maintained by means of circular causality. However, being a general theory of complex systems, Synergetics also deals with phase transitions—the processes associated with qualitative change. More specifically, Synergetics pays special attention and elaborates on the inter-relations between steady state and phase transition. 

FEP is specifically akin to Synergetic 2nd foundation: both consider the mind/brain as an inference device that operates by means of an action-perception play that ongoingly updates the brain’s information: following an update the mind/brain produces series of predictions (internal states) about the environment, and updating them in light of the information that comes from the environment (external states). In other words, top-down predictive models are compared with bottom-up representations by means of embodied action-perception. In this process, the prediction rule for the next step is based on the previously learned Free Energy, or, in terms of Synergetics’ 2nd Foundation, on the potential landscape, *V*. Thus, FE and *V* may be interpreted as “generative models”. The difference between Friston’s FE and synergetics’ *V* is: the learning of *V* is completed or supposedly completed, whereas, in Friston’s approach, it is ongoing. 

### 2.3. Previous Comparison between FEP and SIRNIA

Both FEP and SIRNIA conceptualize a circularly causal play between information constructed in an agents’ mind/brain (*internal states* in the language of FEP and *internal representations* in terms of SIRNIA) versus the information that flows from the environment (*external states* in the language of FEP and *external representations* in terms of SIRNIA). Yet, the FEP ‘states’ and the SIRNIA ‘representations’ differ from each other. The internal representations are mind/brain constructs of the external world, while external representations might take the form of agents’ behavior and action in the environment, but also stand-alone artificial objects the agents produce by means of their bodies, minds, and tools (e.g., buildings, neighborhoods, and whole cities).

Both FEP and SIRNIA employ Shannon’s theory of information [15,16]. In Friston’s FEP, ‘information’ (similarly to ‘free energy’) is a mathematical construct that maintains the system in a steady state, away from phase transition, thus minimizing surprise. In SIRNIA, Shannon information (SHI) explicitly refers to quantity of information when the task of the system is to produce qualitative semantic and pragmatic information (e.g., pattern recognition and action/behavior, respectively). From the latter difference follows a major gap: Friston’s FE is based on a perception-action play that gradually updates the ‘internal state’ in the face of external inflow of information from the environment, thus keeping the system in steady state. In SIRNIA, often, the perception-action play indeed leads to a steady state, however, in other cases, to phase transition. This is so since in SIRNIA semantic and pragmatic information are generated by means of the *inflation* or *deflation* of SHI; in some instances, this process maintains the system is steady state, while, in others, it is associated with a sequence of phase transitions, as illustrated in Figure 2.

A more prominent example concerns the face-girl case of hysteresis (Figure 3): In processes of visual perception, repeated observations may either lead to an improved recognition of an object (in line with Friston’s FEP), or, as in the case of hysteresis, in a first step to saturation, i.e., to fading away of the image to give way to the recognition of another object, that is to say, to a phase transition (Fading away after few seconds happens also when a person’s gaze is fixed. Saccades renew time and again the recognition process.). The result is a kind of dialectics in which the tendency to maintain a steady state and avoid surprise is the very cause of surprise, that is, of phase transition. The whole process is reminiscent of the random walk of a person in the dark in a landscape with one or two attractors. 

To conclude, the FEP emphasizes the role of feedback in the perception-action process: given a certain predictive perception, the associated action feeds back by confirming or correcting the prediction. If it confirms, it reproduces and consolidates the model; otherwise, it corrects and, thus, improves the model. Such a feedback exists also in SIRNIA, though implicitly. On the other hand, SIRNIA emphasizes the play between information inflation and deflation in the process of perception—a play that is missing in FEP. (For a further detailed comparison, see Reference [5] (Chapters 7, 8).

### 2.4. On Schrödinger’s *What Is Life* and Friston’s FEP

Schrödinger’s [4] *What is Life* can be described as a physicist’s view on life: Life, suggested Schrödinger, is delayed entropy; by means of the process of metabolism, an organism—a living system—“… feeds upon *negative entropy*, attracting … a stream of negative entropy upon itself, to compensate the entropy increase it produces by living and thus to maintain itself on a stationary and fairly low entropy level.” This notion of ‘negative entropy’ attracted criticism from physicist colleagues and in response Schrödinger wrote:

“... if I had been law catering for them alone [e.g., physicist colleagues] I should have let the discussion turn on *free energy* instead [italics added]. It is the more familiar notion in this context. But this highly technical term seemed linguistically too near to energy for making the average reader alive to the contrast between the two things. He is likely to take free as more or less an epitheton ornans without much relevance, while, actually, the concept is a rather intricate one, whose relation to Boltzmann’s order-disorder principle is less easy to trace than for entropy and ‘entropy taken with a negative sign’, which by the way is not my invention.”

With his emphasis on life as open system, his notion of *negative entropy* (later termed *negentropy* by Brillouin [19]), his suggestion that “Organization is maintained by extracting ‘order’ from the environment”, and his notion of ‘order from disorder’, Schrödinger’s *What is Life* can be considered one of the forerunners of complexity theory [18]. For example, his ‘order from disorder’ anticipated *order out of chaos* that became a prominent motto of complexity theory. However, while ‘order from disorder’ and his other aforementioned concepts were embraced by the various theories of complexity, the notion of free energy was not and had no significant influence. Recently, however, the notion of free energy re-appeared in Friston’s FEP and in relation to Schrödinger’s question *What is Life.* In a recent study of the Friston’s group, Ramstead et al. [20] have suggested that “the FEP affords a unifying perspective that explains the dynamics of living systems across spatial and temporal scales.” 

As noted above and shown below, the free energy principle to which Schrödinger refers has its origin in the work of Helmholz in thermodynamics and of Feynman in statistical mechanics. Originally, it was a ‘terminus technicus’ to characterize a specific mathematical procedure to calculate an energy, e.g., of a ferromagnet. However, in numerous other cases, including Friston’s, the quantity to be calculated is not an energy, but, e.g., numbers of populations of excited neurons, etc. Nevertheless, the formalism is that used by Feynman. The implication is that taking Schrödinger’s view on complex systems as starting point is too narrow. How can we describe a complex system composed of many interacting parts away from thermal equilibrium, but in steady state? 

### 2.5. The System’s Probability Distribution

In our article, we deal with complex systems, e.g., an animal, brain, city, society, etc. As noted above (Section 2.1), according to Synergetics 1st foundation, a central property of such systems is that their order emerges spontaneously, that is, by means of self-organization. More specifically, the interaction between the parts of the system gives rise to an order parameter that then enslaves the parts, and so on, in circular causality. In order to deal with such systems operationally, thus, it is crucial to identify and define the order parameter(s). But, here, we are facing a difficulty: Because of their complexity, our knowledge on such systems is *inexhaustible* in the sense that we have only incomplete knowledge on their structure, function, and properties. To cope with this uncertainty, i.e., to be able to make predictions, nevertheless, we rely on guesses that are formulated by means of probabilities. More precisely speaking, we are in search for a global probability distribution for the entire system.

The systems we consider are open, i.e., they exchange matter, energy, and information with their surroundings. What are the quantities *P* depends on? They must refer to the properties of the system, per se, but also to the impact of the surround.

In the first place, we aim at a quantitative approach based on *observed* (measured) data on the proper system and its surroundings. (Note that there is the problem of quantification of some qualities, e.g., qualia). Actually, in other approaches, “hidden variables” are also included, e.g., in neural net theories or in Friston’s work. In such cases, contact to observables must be made by specific architectures (e.g., Hinton) or by “generative models” (see below), respectively.

We deal with self-organizing systems. These are systems that acquire their structure or perform functions without direct external “steering”. To model self-organization, two kinds of quantities are considered.

(1)Parameters $\Lambda =\left(\Lambda_{1},\;\ldots \Lambda_{k}\right)$ that are fixed externally (some of them are used as control parameters).(2)Variables $q=\left(q_{1},\;\ldots \;q_{j}\right)\;$that describe the dynamic response of the system to Λ.
In, e.g., our “laser paradigm”, Λ may quantify the energy input into the laser crystal, and *q* the laser light intensity. All in all, *P* is a function of Λ and *q*,
P=P(q;Λ).

The two basic problems are:(1)How to derive P=P(q;Λ)?(2)How to utilize the information contained in P=P(q;Λ)?

To answer (1), we may proceed along two pathways: bottom-up (microscopic approach) or top-down (macroscopic approach). Our microscopic approach based on the laser paradigm helps us to answer the Schrödinger’s “what is life” question: The life process is enabled by an *influx* of (free) energy. Note that the influx *rate* is the crucial parameter and not the free energy, per se. The main part of our paper deals with macroscopic approaches and their interrelations; cf. Figure 4. Thus, we shed light on the derivation of *P*, as well as on its use.

In Section 3, we sketch the microscopic theory of an open system. Thus, the calculated probability distribution *P* will inspire us to formulate Synergetics 2nd Foundation in Section 4.7.

## 3. Microscopic Theory

### 3.1. Goal

In this section, we derive P(q;Λ) of an open complex system by means of microscopic/first principles theory. Our example is laser light [21]. For the sake of completeness, we mention that the underlying theory belongs to quantum mechanics (motion of electrons in atoms), quantum electronics (the electro-magnetic (light-) field)), and the coupling of atoms and field to reservoirs, which act as sources or sinks of energy of the individual components, atoms, and light waves (“open system”). To make our contribution understandable to readers unfamiliar with the strange world of quantum physics, we use a formulation in terms of classical physics, that can be justified by means of the *principle of quantum-classical correspondence*. This principle establishes a one-to-one correspondence between the quantum-mechanical density matrix and the Wigner distribution function so that all quantum-mechanical results can be translated into classical (“c-numbers”) expressions. Having established this, we sketch the steps that lead us to P(q;Λ).

### 3.2. Experimental Set-Up

The heart of a laser (Figure 5) is a rod (of some length L) that contains light-emitting atoms, e.g., atoms positioned in a crystal, such as ruby (its red color stems from those atoms (or ions)). The endfaces of the rod are covered by mirrors, one of them semitransparent so that light can be emitted. The mirrors hold those light-waves that are emitted in axial direction of the rod, so that it can interact with the atoms over a sufficient period of time. The atoms (or ions) are excited by light from lamps surrounding the rod.

### 3.3. Basic Variables, Parameters and Processes

The crucial variable is the electric field strength E(x,t) (perpendicular to the rod’s axis) of the eventually produced laser wave at position x along the rod’s axis at time t. We decompose E into
(1)$$E\left(x,t\right)=q\left(t\right)\exp\left(-2\pi i\nu t\right){\left(hv\right)}^{1/2}\;L^{-1/2}\exp\left(2\pi ix/L\right)\;$$+ Conj. Complex.

q(t) is *the* variable we are interested in to derive P(q;Λ)! All other quantities in (1) are considered fixed (ν: frequency determined by the atomic transition (cf. below), h: Planck’s constant). Note that q is complex, $q=q_{1}+iq_{2}$,$q_{j}$ real.

The field E, or, equivalently, q, is generated by the light-emitting atoms that act like miniature antennas, where electric currents oscillate to generate radiowaves. These “dipole” oscillations occur at the fixed frequency ν as above, but are modulated by a time-dependent variable α(t) so that we use a decomposition analogous to (1), but without the *x*-dependence. We distinguish the atoms by an index μ, so that we consider $\alpha_{\mu }\left(t\right)$, μ=1,2,…,N. N total number of atoms. 

All these little antennas generate the field, i.e., in our representation, q(t), so that
(2)$$\dot{q}\left(t\right)=-ig\;\sum_{\mu }\alpha_{\mu }\left(t\right)\;.$$The dot $\dot{}$ means time derivative.

Where g is a coupling constant between field and atoms, most important, g is of the dimension (1/time). The factor i fixes the phase-relation between $\alpha_{\mu }$ and q. 

Now, we have to take a crucial aspect into account. The laser is an *open* system. In the case of Equation (2), it means that the laser light is emitted from the rod to the surrounding (that has a certain temperature). This means that the field is coupled to a loss-reservoir. Its modeling is a formidable task; here, it must suffice that this coupling has two effects on q:(1)it gives rise to damping—κq,(2)and to a fluctuating force F(t), where the statical average is (These fluctuating forces change very quickly and are sometimes referred to as random fluctuations).

$$\langle F\left(t\right)\rangle =0,\;\langle F\left(t\right)F^{*}\left(t^{\prime }\right)\rangle =Q_{th\;}\delta \left(t-t^{\prime }\right),$$
where “th” refers to “thermal”, and *δ* is Dirac’s function. Thus, all in all, we arrive at our first fundamental equation:(3)$$\dot{q}=-\kappa q-ig\sum_{\mu }\alpha_{\mu }+F\left(t\right)\;.$$Note that $\alpha_{\mu }$ is a time-dependent variable, $\alpha_{\mu }\left(t\right)$. Its time-dependence is determined by an equation that, in analogy to (3), contains three parts:(a)stemming from the interaction of atom *μ* with the field represented by q(t);(b)coupling of atom *μ* to a reservoir leading to damping with a rate constant γ;(c)and to a fluctuating force $\Gamma_{\mu }\left(t\right)$, characterized by

(4)$$\langle \Gamma_{\mu }\left(t\right)\rangle =0,\;\langle \Gamma_{\mu }\left(t\right)\Gamma^{*}{}_{\mu^{\prime }}\left(t^{\prime }\right)\rangle =Q_{\mu }\;\delta_{\mu \mu^{\prime }}\delta \;\left(t-t^{\prime }\right).$$Of particular interest is the form of (a).

As is known from electrodynamics, an oscillating electro (‒ magnetic) field gives rise to an oscillating dipole, which, in our somewhat simplified approach, would lead to an equation:(5)$${\dot{\alpha }}_{\mu }=igq\;.$$Here, we must take a peculiarity into account that is due to quantum theory.

The laser atoms are quantum systems having discrete energy levels. Here, we consider atoms with only two levels, the ground state (with label 1) and the excited level 2, and their corresponding (average) occupation numbers are $N_{1}$ and $N_{2}$. Emission of a light wave by “our” atom means that the difference (called inversion) $d=N_{2}-N_{1}$ goes from +1 to −1, whereas, in case of absorption, d changes from −1 to +1. As some discussion shows, this effect entails that the r.h.s of (5) must be equipped with a factor $d_{\mu }$.

Taking (a)–(c) together, we arrive at our second set of fundamental equations:(6)$${\dot{\alpha }}_{\mu }=-\gamma \alpha_{\mu }+igqd_{\mu }+\Gamma_{\mu }\left(t\right)\;.$$Clearly, $d_{\mu }$ is a time-dependent variable so that we need an equation for it. This equation has the same structure as (6) and is outlined above (a)‒(c). With h: Planck’s constant, ν: frequency, and $h\nu d_{\mu }$ is the energy content of atom μ. The following equation for $d_{\mu }$ (7) can be interpreted as energy balance equation (provided we multiply both sides by hν). The change of energy of atom μ is given by (up to factor hν)
(7)$${\dot{d}}_{\mu }=\gamma_{1}\underset{I}{\left(d_{0}-d_{\mu }\right)}+2ig\underset{\mathrm{II}}{\left(\alpha_{\mu }q^{*}-\alpha^{*}{}_{\mu }q\right)}+\underset{\mathrm{III}}{\Gamma_{\mu d}}.$$
(* means: complex conjugated, i→−i).

The term (II) represents the conversion of energy of atom μ into that of the field: generation of laser light, and has a classical analogue: work done by the field on a dipole. (I) is the energy flux from external energy sources that restore a requisite amount of inversion (~energy of the atom) within a relaxation time $T=1/\gamma_{1}$. (III) is the fluctuations caused by the reservoir (external energy source).

### 3.4. Summary of the Basic Laser Equations

Equations (3), (6) and (7) are, in the present context, the basic laser equations. The variables are q(t), $\alpha_{\mu }\left(t\right)$, $d_{\mu }\left(t\right)$, μ=1,…,N.

Typically, the number N of laser atoms is of order 10^17^ or larger, where each atom by itself is a complicated (quantum) system.

Equations (3), (6) and (7) describe processes, e.g., of energy transfer as (7), when multiplied by hν.

The relevant constants are g, $\gamma_{1}$, γ, all of the dimension (1/time). Most important is the parameter $d_{0}$ which is regulated by the external energy input and serves as control parameter—that the experimenter can adjust.

Now, having a microscopic theory at hand, how can we derive P(q;Λ), thus returning to our goal stated in the beginning of this section?

As Equation (3) reveals, the field q (or E) is generated by the collective action of the atoms ($\underset{\mu }{\sum }\alpha_{\mu }$!), i.e., very many microscopic elements, where q(t) is a directly measurable macroscopic quantity. As close analysis of Equations (3), (6) and (7) shows, we are dealing here with a process of self-organization, where q(t), in the sense of Synergetics, plays the role of an order parameter, and the numerous variables $\alpha_{\mu }$, $d_{\mu }$ can be eliminated by means of the slaving principle. In the present context, the details are not relevant so that we quote the final result:(8)$$\dot{q}=\left(-\kappa +G\right)q-Cq\left(q^{*}q\right)+F_{tot}\;.$$To achieve laser action, the “gain” rate, G, must be larger than the loss rate κ (this is a condition derived by Schawlow and Townes [22]). In our presentation,
(9)$$G=g^{2}D_{0}/\gamma ,\;D_{0}=Nd_{0}\;.$$C is a “saturation” constant,
(10)$$C=4g^{4}D_{0}/\gamma^{2}\gamma_{1\;}\;.$$Note that κ, *G*, *C* have the dimension (1/time).

The stochastic properties of $F_{tot}$ (the total fluctuation) are characterized by
$$\langle F^{*}{}_{tot}\left(t\right)\rangle =\langle F_{tot}\left(t\right)\rangle =0$$
(11)$$\langle F^{*}{}_{tot}\left(t\right)F_{tot}\left(t^{\prime }\right)\rangle =2\kappa \left(n_{th}+n_{sp}\right)\;\delta \;\left(t-t^{\prime }\right)=Q_{tot}\;\delta \;\left(t-t^{\prime }\right)\;,$$
where $n_{th}$ is the number of photons at temperature T, and $n_{sp}$ the number of spontaneously emitted photons. In what follows, we may treat $n_{th}$ and $n_{sp}$ as fixed parameters.

Equation (8) is of the type of Langevin equation, which describes a process that has:(a)a deterministic cause: the first two brackets in (8);(b)a stochastic cause, $F_{tot}$.

Now, we can implement the most important step.

### 3.5. Derivation of the Probability Distribution P

This is achieved by converting an equation of the Langevin type (cf. (a) and (b) above) into a Fokker-Planck equation. Just as a reminder and for illustration: let the Langevin equation for a real variable x(t) be:

(12)$$\dot{x}=-\gamma x+F\left(t\right)\;,$$(13)$$\langle F\left(t\right)\rangle =0,\;\langle F\left(t\right)F\left(t^{\prime }\right)\rangle =Q\;\delta \;\left(t-t^{\prime }\right)\;.$$The corresponding Fokker-Planck equation of the time-dependent probability distribution function *P*(x;t) reads
(14)$$\dot{P}=-\frac{\partial }{\partial x}\left(-\gamma xP\right)+\frac{1}{2}Q\frac{\partial^{2}}{\partial x^{2}}\;P\;.$$Its steady state solution reads $\dot{P}=0$:(15)$$\dot{P}\left(x\right)=N\exp(-\frac{\gamma x^{2}}{Q})\;.$$N: normalization factor.

We skip the Fokker-Planck equation belonging to (8) and immediately write down its steady state solution, which is the probability distribution *P*,
(16)$$P\left(q\right)=N\exp\left(\frac{1}{Q_{tot}}\left\{\frac{1}{2}\left(G-\kappa \right)\left(q^{*}q\right)-\frac{1}{4}C{\left(q^{*}q\right)}^{2}\right\}\right)\;.$$$Q_{tot}$ is the total fluctuation intensity. * This distribution function can be linked to the photon distribution function that has been precisely measured, confirming (16).

The constants $Q_{tot}$, G, C are defined in (11), (9), (10), respectively, and are all of the same dimension (1/time) so that the exponent of (16) is dimensionless as it must be.

When, in (16), we multiply numerator and denominator by hν, the exponent acquires the form:$$\frac{energy\;flux\;due\;to\;dynamics}{energy\;flux\;due\;to\;noise}.$$The importance of our results lies in the fact that it allows the comparison between P of an *open* system, away from thermal equilibrium, with that of a system in thermal equilibrium, e.g., a ferromagnet (see below). Here, P can also be expressed by an exponential function for which the exponent is written as ratio, but this time as
(17)$$-\frac{1}{kT}free\;energy\;.$$k: Boltzmann’s constant, T: absolute temperature.

Thus, though *P* (nonequilibrium) and P (equilibrium) have the same mathematical form, the meaning of the quantities and their dimensions are quite different. As we will show below, this difference has far-reaching consequences on the application of general principles, such as Maximum Entropy, Free Energy Minimization, Friston’s approach to biology, e.g., action/perception, Feynman’s Free Energy approach, and so on (see Figure 4).

## 4. The Free Energy Principle and Its Metamorphosis: From Helmholtz (Thermodynamics) and Feynman (Statistical Mechanics) to MacKay (Information Theory) and Friston (Life Sciences)

### 4.1. Helmholtz Free Energy

The Free Energy Principle has its roots in thermodynamics, the theory of heat. It deals with relations between measurable quantities of physical objects, e.g., a gas in a volume, a ferromagnet, a liquid, etc. Typical quantities are energy (e.g., mechanical: kinetic and potential energy), heat as a form of energy E, temperature T, and less directly accessible to our senses, entropy, S. The introduction of this quantity is necessitated by our observation of relaxation processes/irreversibility. When we bring two bodies at different temperatures in contact, heat flows from the warmer to the colder body until both acquire the same temperature. The reverse process is never observed. This is reflected by the principle that, in a closed system, the entropy *S* can never decrease. S can be measured, and its dimension is energy/temperature. In a nutshell: the free energy F is defined by Helmholtz [7] as
(18)F=E−TS ,
where E is the “inner energy”. *F* is that maximum amount of energy of a physical object (say gas in a tube) that can be converted into mechanical work, e.g., shifting the piston of the tube (cylinder) (“bottle”), against an external force. According to thermodynamics, F of a closed system acquires a minimum value.

### 4.2. Thermodynamics and Information Theory Have the Same Root: Combinatorics and Large Numbers

*Thermodynamics* is a phenomenological, macroscopic theory: it does not care what matter consists of. This changed when Boltzmann [23] took into account that matter, e.g., a gas or a piece of metal, is composed of atoms or molecules. To make a long story short: Planck [24] cast Boltzmann’s result in the form
(19)S=klnW ,
where S is the entropy, k Boltzmann’s constant, and W is the number of “microstates” (properly normalized; cf. Reference [24]). For illustration: consider a chain of N molecules of the sort 1 and 2 with numbers $N_{1}$, $N_{2}$, respectively, with $N=N_{1}+N_{2}$. 

Then, there are
(20)$$W=\frac{N!}{N_{1}!\;N_{2}!}\;$$
different chains. According to Boltzmann [23], such macrostates occur that can be realized by a maximum of W microstates:maximum of entropy S (19) !

For practical application of S, we need a handier expression. First, we note that “properly normalized” means that W must be divided by the total number of atoms, and the limit $N_{1},\;N_{2},\;N\to \infty$ must be taken. By use of Stirling’s formula in the approximation
(21)lnN!≈N(lnN−1) ,
we may cast (19) in the form
(22)$$S=k\left(-P_{1}\ln P_{1}-P_{2}\ln P_{2}\right),\;$$
$$Pj=\frac{Nj}{N}\;for\;N\to \infty .$$Pj is the relative frequency that atom j (=1 or 2) occurs. When there are j different types of atoms, then (22) generalizes to the well-known formula for Gibbs entropy:(23)$$S=-k\sum_{j=1}^{j}P_{j}\ln P_{j}\;.$$Let us turn to the concept/definition of *information* as defined by Shannon [15,16]. How many messages per second can be sent through a “channel”, e.g., a cable? A message is represented/encoded by a specific sequence of symbols, such as dashes and dots of the Morse alphabet. How many different sentences of N symbols with $N_{1}$ dashes and $N_{2}$ dots can we form? Obviously, W(20). Since the transmission of each symbol takes some time, the larger the channel capacity is, the larger $W^{\prime }=W/N$ is. Thus, a measure of Shannon information depends on $W^{\prime }$. In a last step, following Shannon, we require that Shannon information is defined by
(24)$$I=K\ln W^{\prime }\;,$$
where K is a constant. In complete analogy to the thermodynamic case, we consider $W^{\prime }$ in the limit N→∞ that leads us to (22) with k replaced by K, or, in case of J symbols, to (23, k→K). The only difference between Boltzmann and Shannon entropies consists in the different factors k and K, respectively. To obtain “Shannon”, we choose K such that K ln becomes log_2_ so that, finally,
(25)$$I=-\sum_{j=1}^{j}\;P_{j}\log_{2}P_{j}\;.$$This entails that I is measured in bits (i.e., number of yes/no decisions). In the following, we write *P*(*q*) instead of *P_j_* when *j* is replaced by a set of indices (cf. Equation (26)).

### 4.3. Jaynes’ Maximum (Information) Entropy Principle

This principle allows us to make the best guess on a complex system of which only a limited set of data is known. In its original form, Jaynes [25,26] applied it to thermodynamics, i.e., systems in thermal equilibrium. We present this approach by means of an explicit example: the ferromagnet. It is a complex system: on the microscopic level, it is composed of tiny magnets (“spins” with “magnetic moment”) situated on a three-dimensional lattice. These magnets may point in only two directions, “up” and “down”, and have the same size (“magnetic moment”). To describe a microstate, we label the elementary magnets by l=1,…, N. 

N: total number. A specific microstate is characterized by a “state vector”:(26)$$q=\left(q_{1},\;q_{2},\ldots ,q_{N}\right),\;\mathrm{where}\;q_{l}=+1,\;or-1\;.$$Since the magnets interact with each other, with each configuration represented by q, an energy E(q)  is connected (For an explicit example, see below, Section 4.5).

Finally, because the ferromagnet has some temperature (it is assumed to be in thermal equilibrium), its elementary magnets flip randomly between up and down. Thus, the states q obey a temperature-dependent probability distribution P(q;T). Once we know *P*, we can calculate all macroscopic quantities, such as the total energy of the ferromagnet, the magnetic field it produces, etc. 

Jaynes’ principle assumes (in accordance with Boltzmann’s principle) that those macrostates occur that can be realized by a maximum number of microstates (23). The number of microstates has been calculated above and appears in Boltzmann’s or Shannon’s Formulas (6) and (8), respectively—up to different constant factors k, K, or the use of *ln* or log_2_. 

While Jaynes used Shannon entropy, thus, referring to information theory, here, we start right away from the thermodynamic form (6).

The crucial point is that not all configurations q are equally probable but are *constrained* by the condition that they give rise to the macroscopically measured (or measurable) quantities, such as the total energy. These measurable quantities $f_{j},\;j=1,\ldots ,J$, are linked to P(q):(27)$$f_{j}=\sum_{q}P\left(q\right)f_{j}\left(q\right)\;,$$
where, e.g., $f_{1}$ is the expression for the energy E(q) of the configuration q. These considerations lead us in a straightforward way to Jaynes’ principle: to derive P(q) by maximizing the number of microstates S, as expressed by (23) under the constraints (27) and the normalization condition:(28)$$\sum_{q}P\left(q\right)=1\;.$$This task is solved by means of Lagrange multipliers Λ.

We obtain
(29)$$P\left(q\right)=\exp\left(\Lambda_{0}-\sum_{j}\Lambda_{j}f_{j}\left(q\right)\right)\;.$$For systems in thermal equilibrium, one constraint is always total energy so that, typically,
(30)$$P\left(q\right)=\exp\left(\Lambda_{0}\right)\;\exp\left(-\Lambda E\left(q\right)\right)\;,$$
which is just the Boltzmann distribution function, so that we can identify
$$\exp\Lambda_{0}=N,N\;\mathrm{normalization}$$
Λ=1/kT, k= Boltzmann’s constant, T=absolute temperature.

The relations (29) or (30) enfold still more important relations.

Inserting (30) in (28) yields
(31)$$\exp\left(-\Lambda_{0}\right)=\sum_{q}\left(-E\left(q\right)/kT\right)\;\equiv Z\;,$$
where *Z* is the partition function. Knowing it, we can calculate all thermodynamic quantities, such as total energy at T, the magnetic field of a ferromagnet at T, etc. (e.g., putting β=1/kT, the total energy is given by $-\frac{\partial }{\partial \beta }\ln Z$). When we insert (30) in (23), we find, using (32),
(32)$$S=-k\Lambda_{0}+\frac{1}{T}\;\sum_{q}E\left(q\right)P\left(q\right)\;,$$
(33)$$\underset{q}{\sum }E\left(q\right)P\left(q\right)=\langle E\rangle$$
is the energy of the system at temperature T. Rearrangement of (32) yields
(34)$$k\Lambda_{0}=\frac{1}{T}\langle E\rangle -S\;.$$To reveal the meaning of $\Lambda_{0}$, we put, in (34),
(35)$$\Lambda_{0}=\frac{F}{kT}\;$$
and obtain
(36)F=〈E〉−TS ,
that is just the expression of Helmholtz Free Energy! This *macroscopic* relation is brought about via a microscopic theory!

As stated after (31), knowing Z, we can calculate important macroscopic quantities. But, according to (31), Z can be expressed by $\Lambda_{0}$, which, in turn, can be expressed by the free energy so that, finally,
(37)F=−kTlnZ .This reveals the central role played by F. So, the crucial question arises:

How can we calculate F by means of the microscopic theory? For instance, in the case of the ferromagnet, this problem turns out to be extremely difficult; it is here where Feynman’s principle comes in.

### 4.4. Feynman’s Free Energy Principle

Since, in important cases, the calculation of F by means of an exact microscopic theory is a “hard” problem, the question arises: how far can we calculate F approximately? 

To answer this question, we recall the definition of F(1) in terms of the microscopic expression for E (33) and S (23),
(38)$$F=\sum_{q}E\left(q\right)P\left(q\right)-kT\sum_{q}P\left(q\right)\ln P\left(q\right)\;.$$We obtain an approximate expression for F, $\tilde{F}$, when, in (38), we replace P(q) by another probability distribution, Q(q), we can (more) easily handle:(39)$$\tilde{F}=\sum_{q}E\left(q\right)Q\left(q\right)-kT\sum_{q}Q\left(q\right)\ln Q\left(q\right)\;.$$To get an insight into the quality of our approximation, we derive a suitable expression for $\tilde{F}-F$. Replacing E(q) in (38), using $P\left(q\right)=Z^{-1}\exp\left(-E\left(q\right)/kT\right)$, we arrive at:(40)$$\tilde{F}-F=kT\;D_{kl}\left(Q||P\right)\;,$$
where
(41)$$D_{kl}\left(Q||P\right)=\underset{q}{\sum }Q\left(q\right)\ln\frac{Q\left(q\right)}{P\left(q\right)}\;$$
is the Kullback-Leibler divergence. It has the crucial property
(42)$$D_{kl}\geq 0,\;(“\mathrm{Gibbs}\;\mathrm{inequality}”),$$
(43)=0 only if P(q)=Q(q) .This means that any approximate free energy $\tilde{F}$ is always larger than the true free energy F.

This is the basis of Feynman’s Free Energy Principle [8].

In practical applications, Q is chosen in form of an explicit function with adjustable parameters so that $\tilde{F}$ becomes minimized. Note that the primary goal is to achieve a good approximation to the true probability distribution P(q).

### 4.5. An Example: The Ising Model

This is the model of a ferromagnet in a homogeneous magnet field. The energy function can be written in the form:(44)$$E\left(q\right)=-\frac{1}{2}\;\sum_{mn}J_{mn}q_{m}\;q_{n}-\sum_{h}hq_{n}\;.$$Each $q_{m}$ can acquire only the value +1 (“spin up”) or −1 (“spin down”). $J_{mn}$ represents the interaction energy between spins (or elementary magnets) at lattice sites labeled by m (or n). It is assumed that $J_{mn}=J>0$ for neighboring lattice sites, and =0 otherwise. To calculate the approximate free energy (39), Q is chosen as
(45)$$Q\left(q\right)=\frac{1}{Z_{Q}}\exp\left(\sum_{n}a_{n}q_{n}\right)\;,$$
i.e., as a product of functions each containing only one variable, $q_{n}$.

The constants $a_{n}$ are variational parameters to be chosen such that $\tilde{F}$ is minimized. As a little analysis shows, the expressions for E and S in (39) can be explicitly calculated, yielding simple expressions of $a_{n}$. Because of the equal role of all spins, $a_{n}=a$ is the only variational parameter.

The exponent of exp in (45) is an example of a typical hypothesis on Q; we will discuss more examples below. The product entails that each individual spin is subject to a mean field generated by all other spins. This “Mean field approximation” ignores correlations among the individual spins.

### 4.6. The Free Energy Principle beyond Physics

The FEP owes its designation to Equation (38), which is of the form:(46)free energy=energy−const x entropy .In (39), energy appears as average (33) (“expectation value” of an energy expression, such as (44)), and entropy as presented in (23) and (25) is, up to the factors k or K, nothing but a measure for the amount of microstates. The relative weight with which energy and entropy enter (39) is temperature, T. While maximal entropy requires that all microstates (“configurations”) are realizable and equally important, the energy minimization prefers some microstates over others (Think of a pearl necklace with an equal number of white and black pearls. Then, a necklace where all white pearls are linked together may be more boring (“less priced”) than one where the white and black pearls appear in an altering sequence.). The FE principle seeks a compromise by minimizing (46). This kind of balance principle holds for many systems, e.g., biological, economic, etc.

Depending on the application, the energy expression may be replaced by the one of interest, e.g., number of sequelae. 

In general, the typical entropy expression (23) without the factor k may be invoked, but there may be also other “cost” functions.

A first application of the free energy principle to a purely information theoretical problem outside the natural sciences is due to MacKay [9,14]. This underlines that this principle is a purely mathematical approach.

### 4.7. Synergetics 2nd Foundation

As we have shown in Section 4.3, Jaynes’ Maximum (Information) Entropy Principle [25,26] allows a straightforward derivation of fundamental relations of thermodynamics. This special application rests on the use of *energy* as constraint, which is assumed to be a (measured) time-independent, fixed quantity. But this assumption is no more justified when we deal with open systems, which exchange *energy*, matter, and/or information with their surroundings. So, the basic question arises of whether Jaynes’ principle can be applied to open systems, provided we use other constraints than energy. A first hint comes from the microscopic theory of an open system in which the prototype is the laser. We have found an explicit expression for the probability distribution of the laser field amplitude q (16) of Section 2 that is of the type
(47)$$P\left(q\right)=N\exp\;\left(aq^{2}-bq^{4}\right)\;$$(here, for simplicity, q: real). Which constraint(s) do we need to derive (47)? As a glance at the general formulation of Jaynes’ principle (29) of Section 4.3 reveals, there are two constraints (besides normalization), namely the moments
(48)$$\langle q^{2}\rangle ,\;\langle q^{4}\rangle ,\;$$
where
(49)$$\langle q^{m}\rangle =\int P\left(q\right)q^{m}dq\;.$$When there are several variables, $q_{1},\ldots ,\;q_{L}$, involved, the constraints (48) can be generalized to polynomials of some order. Most important expressions are correlation functions,
(50)$$\langle q_{l}\;q_{k}\rangle \;,$$
which can be linked to the variation $\nu \left(q_{l,}q_{k}\right)$:(51)$$\nu \left(q_{l},q_{k}\right)=\langle q_{l}q_{k}\rangle -\langle q_{l}\rangle \langle q_{k}\rangle \;.$$All in all, Synergetics 2nd Foundation is an application of Jaynes’ principle where the constraints are polynomials that appear in the function $f_{j}\left(q\right)$ in the exponent of (29) of Section 4.3. If the polynomials are of first and second order, (29) becomes a Gaussian which allows us to evaluate the constraints 〈P(q)f(q)〉 explicitly. On the other hand, (29) of Section 4.3 as Gaussian cannot represent multistable states—this requires the inclusion of 4th order terms (see, e.g., Haken and Portugali in Reference [5]), where pattern recognition is dealt with.

A major task may be the calculation of the Lagrange parameters in (29) of Section 4.3. While, in the Gaussian case (linear and quadratic constraints), this problem can be reduced to linear algebra (see, e.g., Haken and Portugali [5]), approximation methods must be invoked in the general case. An explicit example is this: Let the true, but unknown, probability distribution be given by P(q). We try to approximate it by
(52)$$\tilde{P}\left(q;\lambda \right)=\tilde{Z}{\left(q;\lambda \right)}^{-1}\exp\left(-\sum_{j}\lambda_{j}V_{j}\left(q\right)\right)\;,$$
where, in the present context the functions, $V_{j}$ are polynomials. To find the best fit between P and $\tilde{P}$, we use the method of steepest descent, i.e., we try to lower the Kullback-Leibler divergence $D_{kl}$ stepwise. We introduce an iteration parameter that we might interpret as time t, so that, formally, the steepest descent can be written as
(53)$$\frac{d\lambda_{j}\left(t\right)}{dt}=-\gamma \;{\mathrm{grad}}_{j}D_{kl}\left(\lambda \right),$$
where γ is a parameter related to the stepsize.

Inserting P and (52) in $D_{kl}\left(41\right)$ (where $Q\to P,\;P\to \tilde{P}$) and rearranging terms, we readily obtain
(54)$$\frac{d\lambda_{j}\left(t\right)}{dt}=-\gamma \;\left(\langle {\tilde{V}}_{j}\rangle {}_{P}-\langle {\tilde{V}}_{j}\rangle {}_{\tilde{P}}\right)\;.$$In (54), $\langle {\tilde{V}}_{j}\rangle {}_{\tilde{P}}$ is the expectation value of $V_{j}$ that can be measured, while $\langle {\tilde{V}}_{j}\rangle {}_{P}$ must be calculated. For the derivation of (54), we need to recall
(55)$$\frac{\partial }{\partial \lambda_{j}}\;\ln\tilde{Z}=\langle {\tilde{V}}_{j}\rangle {}_{P}\;.$$

### 4.8. Summary 

All the above relations are based on an application of Jaynes’ MIE Principle that may be considered a consequent mathematization of Boltzmann’s principle: study all microstates that are compatible with constraints. There is, however, another source of inspiration of how to deal with complex systems: Bayesian inference, that we will study next.

## 5. Bayesian Inference

We consider an example of daily life: a traffic light for pedestrians with colors red and green implying “stop” and “go”, respectively. Conventionally, when a pedestrian does not stop, we infer a “green light”. However, a certain (in principle, measurable) percentage of pedestrians go on in spite of “red”. When we see a pedestrian going, how sure can we be that the traffic light was green? To put it quantitively, how large is the *probability* that the light was green?

Another typical problem of neuroscience: a sensory neuron fires when it receives input from yellow light. But it may also fire when the input stems from green light. We observe that neuron firing. With which probability does the light stem from a green source?

These are typical questions that can be answered by Bayesian inference that is based on Bayes’ [11] theorem. This rests on a mathematical tautology. Let the joint probability of two variables r and s be P(r,s). (r: pedestrian walk/go; s: traffic light green/red). Then, we can write
(56)P(r,s)=P(r|s) P(s) ,
(57)=P(s|r) P(r) ,
where, e.g., P(r|s) is the conditional probability that “r” happens “under the hypothesis” that “s” happens. Equating the right-hand sides of (56) and (57) and rearranging terms leads us to
(58)$$P\left(s|r\right)=\frac{P\left(r|s\right)\;P\left(s\right)}{P\left(r\right)}\;.$$We may eliminate P(r) because of probability theory
(59)$$P\left(r\right)=\sum_{s}P\left(r,s\right)=\sum_{s}P\left(r|s\right)P\left(s\right)\;.$$So, that, finally,
(60)$$P\left(s|r\right)=\frac{P\left(r|s\right)\;P\left(s\right)}{\;\sum_{s}P\left(r|s\right)\;P\left(s\right)}\;.$$This formula allows us to answer our above questions.

P(s|r) means: when we observe a specific “reaction” r (e.g., walks), what is the probability of a “signal” s, e.g., green? The “posterior” P(s|r) can be calculated by means of the right-hand side of (60), provided we know (have measured) the conditional probability P(r|s) and prior distribution function P(s).

In our examples, r and s can be represented by discrete variables, e.g., 0 and 1. P(s|r) can be plotted as a matrix with elements $M_{rs}$. So far, the formalism holds for any set of discrete variables. A major extension results when the variables r, s are continuous.

In such a case, a “generative model” for the joint probability P(r,s) is required. To get some insight into such a model, consider a sensory neuron s and an action neuron r. r and s are bidirectionally coupled. s receives inputs from an external source $s_{0}$. The variables r and s represent neural activities (e.g., measured in firing rates). In many practical applications (for computational reasons), the generative model is chosen as a Gaussian. Because of the relations between $r,\;s,\;s_{0}$, it has the form
(61)$$P\left(r,s\right)=N\;\exp\left(-\alpha r^{2}-2\beta rs-\gamma s^{2}-ss_{0}\right)\;,$$
where $\alpha ,\beta ,\gamma ,s_{0}$ are parameters that must be fixed by (fitting) experimental data. 

To this end, the approach described by Equations (53)–(55) can be used. In the last step, the desired posterior can be calculated by means of Bayes’ theorem (60).

### 5.1. Interlude: How to Find a Generative Model?

Our above formulation of a generative model may seem rather arbitrary, an essential motivation is the computational advantage offered by the use of Gaussians—often known as the Laplace assumption in Bayesian statistics [27]. There is, however, a much more profound guide to arrive at a (the) generative model, based on Synergetics’ 2nd Foundation. There we use, in Jaynes’ Maximum Information Principle, moments [25,26] and correlation functions as constraints.

On the other hand, the exploration of correlations lies at the heart of Bayesian inference. How does the behavior of pedestrians correlate with traffic light? How does the activity of action neurons correlate with that of sensory neurons? According to Section 4.7, Synergetics’ 2nd Foundation, the relevant probability distribution is given by (29), Section 4.3, where the functions f(q) are polynomials of the variables, e.g., r and s. In this way, we are quite naturally led to the formulation of a generative model, where the Lagrange parameters are the free parameters. In many cases, it suffices to use polynomials of first and second order, e.g., (61). 

The whole procedure requires that the constraints have to be sufficiently exactly fixed (sufficiently many observations). We recall that, in the case of Gaussian, the $\Lambda_{s}$ can be calculated by linear algebra (cf., e.g., Reference [5]).

### 5.2. Variational Bayesian Inference

As we have seen above, Bayesian inference allows us, via Formula (60), to calculate the posterior, P(s|r), in our notation/examples. In practice, an alternative procedure has turned out to be more “easily” feasible. In this case, one tries to directly approximate P(s|r) by means of a test function Q and to check the quality of the approximation. To explain the procedure, we specialize (3) to the case of specific value $r=r_{m}$ so that
(62)$$P\left(s|r_{m}\right)=\frac{P\left(r_{m}|s\right)P\left(s\right)}{P\left(r_{m}\right)}\equiv \frac{P\left(r_{m},\;s\right)}{P\left(r_{m}\right)}\;.$$In what follows, we consider $r_{m}$ and, thus, $P\left(r_{m}\right)$ as fixed parameters. Thus, Q is also a function of s only (for each parameter value $r_{m}$). 

A quality measure for Q(s) is the Kullback-Leibler divergence
(63)$$\sum_{s}Q\left(s\right)\ln\frac{Q\left(s\right)}{P\left(s|r_{m}\right)}\;,$$
which, due to (62), can be written as
(64)$$\left(63\right)=\ln P\left(r_{m}\right)+\sum_{s}Q\left(s\right)\frac{\ln Q\left(s\right)}{P\left(r_{m},\;s\right)}\;.$$This means that the Kullback-Leibler divergence between Q(s) and $P\left(s|r_{m}\right)$ (63) and the Kullback-Leibler divergence between Q(s) and $P\left(r_{m},\;s\right)$ (64) differ only by a constant. This entails that (63) and
(65)$$\underset{s}{\sum }Q\left(s\right)\ln\frac{Q\left(s\right)}{P\left(r_{m},s\right)}\;$$
attain their minima at the same Q(s). Thus,
(66)$$Q_{min}\left(s\right)=P\left(s|r_{m}\right)\;,$$
where $Q_{min}$ minimizes (65).

This is the variational Bayesian inference.

If s is a continuous variable, the sum over s must be replaced by an integral.

For more details, cf. MacKay [10], and, for a comparison between the free energy principle and Bayesian inference, cf. Gottwald and Brown [28].

## 6. Life as a Steady State

In biology, there are three fundamental questions.


How did life originate?How does it evolve?How does it maintain a steady state, be it at the level of phylogenesis or ontogenesis?


1. is still an unsolved mystery. 

2. is a decisive answer that has been given by Darwin.

3. is the object of “autopoiesis”, a field founded by Maturana and Varela [29]. More recently, the maintenance of life mainly on the ontological level, or *homeostasis* in other words, has been dealt with by Friston from a fundamental perspective based on a Free Energy Principle. 

### 6.1. Friston’s Free Energy Principle. An Example

Friston and his coworkers have dealt with his principle in a series of papers [3,30,31]. In the following, we scrutinize an important application of his approach, mainly by means of “his” example of a perception-action cycle. At its bottom lies a model of the action of an animal to maintain its homeostasis. The “brain” consists of a (group of) sensory neuron (*s*) with activity (firing rate) *s*, and bidirectionally coupled to a (group of) action neuron (*s*) with activity *r*. In this model, the action of the environment on the sensory neuron is represented by means of a “hidden” variable ψ. A sensation *s* is associated with food. An action *r* evokes a response ψ of the environment with a conditional probability *P*(ψ|r). This is achieved best if *P*(ψ|r) matches P(ψ|s) best. To this end, we use the Kullback-Leibler divergence and identify in (63) of Section 5.2
(67)Q(s)→ P(ψ|r),
(68)$$P\left(s|r_{m}\right)\to P\left(\psi |s\right).\;$$

Invoking variational Bayesian inference, we have to minimize (65) of Section 5, where, in addition to (67), (68), we replace $P\left(s|r_{m}\right)$ by P(ψ|s). Because of P(ψ,s)=P(ψ|s)P(s), finally, Friston’s Free Energy expression [3] takes the form
(69)F(r,s)=∫dψP(ψ|r) ln(P(ψ|r)/ P(ψ|s))dψ−lnP(s) .Quite remarkably, none of the probabilities in (69) can be measured directly. The relations between r,s, ψ come to light only with help of a generative model. In our book, Haken and Portugali, 2021, we used the model
(70)$$V\left(r,s,\psi \right)=r^{2}-2\beta rs+s^{2}-2s\psi +a{\left(\psi -s_{0}\right)}^{2}\;$$
as example, where β,a, and $s_{0}$ are free parameters. Because of the quadratic form (70), F(69) can be explicitly calculated. For a≫1, but finite, we obtain
(71)$$F\left(r,s\right)=Const.+\frac{1}{a}\;{\left(s_{0}+\beta r-s\right)}^{2}+\left(1-\beta^{2}\right){\left(s-\frac{s_{0}}{1-\beta^{2}}\right)}^{2}\;.$$Since F is the sum of two non-negative functions, F acquires its minimum if
(72)$${\left(s_{0}+\beta r-s\right)}^{2}=0\;,$$
and
(73)$${\left(s-\frac{s_{0}}{1-\beta^{2}}\right)}^{2}=0\;.$$To discuss the significance of (72), (73), we recall that $s_{0},\;\beta$ are parameters, while s and r are variables. What does this mean for an animal’s behavior?(1)A learning phase: by choosing *r* and measuring s, it can determine $s_{0}$ and β (with limits!)(2)A prediction phase: Having fixed $s_{0},\;\beta$ (which happens at the neuronal level), the animal may predict s when it has selected action r.

Our example explicitly shows that the optimal action strategy to reach the minimum of *F* is just to “go downhill” (steepest decent; also cf. Reference [32]).
(74)$$\frac{dr}{dt}=-\frac{\partial F}{\partial r}.$$

In a number of practical cases, however, this procedure is not executed. Think of the marshmallow effect (gratification of children) or of a shift of food uptake by animals in favor of an expected bigger prey. For a recent experiment with cuttlefish, cf. Reference [33].

Quite generally, the free energy function can be visualized as a mountainous landscape with hills and valleys. Ideally, this landscape has a Janus face: either we consider the parameters fixed, or the variables (measured).

In the learning phase, we consider the variables (technically “data”) as given, and the task is to find those parameter values that minimize F, i.e., the “coordinates” of a valley. This may be, in contrast to our simple example, a demanding task. The free energy principle, per se, does not provide us with instructions about the sequence of the individual steps. Humans or animals may start from the assumption of fixed parameters (e.g., $s_{0}$, β), choose a value of r, predict some s, compare it with the actually measured s, and correct the error made by a change of $s_{0}$ and/or β. With improved $s_{0}$, β, they may, in a next step, continue. But the sequence of these steps is beyond the FEP instructions. The selection of the step sequence may require further criteria (depending on previous experience). Quite generally, for explicit calculation, an explicit form of the generative model is needed. As long as it is “Gaussian”, all integrals can be expressed via linear algebra, and the determination of the (Lagrange) parameters be left to the solution of algebraic equations. However, Gaussian distributions have only one maximum, whereas, in reality, the probability distributions may be multimodular (multistable states). Such cases have been studied, e.g., in approaches to pattern recognition (see e.g., Reference [5]). In the context of Friston’s FEP, multistability has been studied recently by Da Costa et al. [12]. 

“The above formulation of free energy minimization is often cast as prediction error minimization [34,35,36]. This is especially true for generative models with additive Gaussian noise. In this setting, the mechanics we have been describing reduces to predictive coding or Kalman filtering. The focus on the prediction error minimization—in the cognitive neurosciences (and beyond)—is fully licensed by the general observation that the gradients of free energy can always be expressed as a prediction error of one form or another, namely the difference between some sampled sensory data and that predicted under a generative model. In short, minimizing free energy can be construed as minimizing prediction errors. In turn, this casts self-organization as the ability of certain systems to destroy the free energy gradients that created them [37]. This Synergetics perspective was, in fact, one of the inspirations for the free energy principle” (Friston K. (Welcome Trust, London), personal communication, 2021).

In some papers, such as Da Costa et al. (ibid), instead of the values of single neurons (firing rates), the mean values of whole neural populations are considered. An early detailed approach dealing with the dynamics of excitory and inhibitory neurons was formulated by Wilson and Cowan [38,39]. For later work, cf., e.g., Reference [40]. 

### 6.2. The FEP and Synergetics 2nd Foundation: A Comparison

Both approaches use (in a nutshell):a variational principle;a generative model.(a)In Friston’s case, the variational principle is FEP, and the formulation of the generative model is in the hands of the “modeler”.(b)In Synergetics 2nd Foundation, the variational principle is Jaynes’ Maximum (Information) Entropy Principle. The formulation of the generative model may be reduced to the selection of constraints in the form of moments and correlation functions, but it can also be chosen freely depending on appropriate constraints. 

When expressed like this, one can see that there is an intimate relationship between the FEP and Synergetics: in the sense that minimizing free energy is just an expression of the maximum entropy principle under constraints. In the Synergetics formulation, these constraints are supplied in the form of Lagrange multipliers that inherit from the physics of the problem at hand. In the free energy principle, these constraints are expressed in terms of prior constraints under a Bayesian generative model. In both instances, the imperative is to provide an accurate account of—usually sparse—data, while maintaining the maximum entropy of that account (i.e., maximizing the entropy of an approximate posterior density). One could conceive of the maximum entropy principle as furnishing a unique solution to an inference or measurement problem in the spirit of Occam’s principle.

Aside from mathematical details, (a) and (b) differ in their basic preconditions. Namely, (b) assumes that the collection of data (measured values of variables) is terminated, whereas (a) deals (at least in principle) with the stepwise data acquisition.

If the generative models of (a) and (b) coincide, in the limiting case of sufficient data, the (Lagrange) parameters are expected to acquire the same numerical values. For an explicit example, cf. Haken and Portugali [5].

#### Exploration of a Room by Blindfolded Persons

The notion of SIRN (and, thus, SIRNIA—see Section 2 above) was originally suggested as a theory regarding the construction of internal spatial representation (e.g., a cognitive map) by means of action-perception—in terms of SIRNIA, by means of the interaction between internal and external representations. A basic feature of a cognitive map is that usually the environment concerned (e.g., a maze or a city) is large to the extent that it cannot be captured by means of a single visual act as in pattern recognition of small-scale objects (e.g., a face). Consequently, the internal representation (cognitive map) is constructed in the brain sequentially as the agent moves in, explores the environment, and accumulates information about it. Movement in the environment is also the basis of the phenomenon of exploratory behavior that refers to the innate tendency of animals to perform a highly structured pattern of exploratory movements when introduced to a new environments [41]. Thus, Yaski, Portugali, and Eilam [42] suggested a link between the processes of cognitive mapping and exploratory behavior. As part of their attempt to explore the role of exploratory behavior in the cognitive mapping process, they conducted a set of experiment with rats [43,44,45] and one with human subjects [42]. In the latter experiment, 10 blindfolded adult human subjects (5 males and 5 females) were introduced into an unfamiliar room, where they were asked to move incessantly for 10 min. Observing the locomotor activity of the subjects, five sequential exploratory behavior patterns were exposed: (1) ‘looping’, (2) ‘wall-following’, (3) ‘step-counting’, (4) ‘cross-cutting’, and, finally, (5) ‘free traveling’. Figure 6 illustrates a typical such process performed by one of the 10 subjects. Similar exploratory behavior patterns were performed by all subjects. After the experiment, the subjects filled a questionnaire from which it was found that all 10 have correctly described the environment as rectangular, 4 have correctly estimated the size of the room, 8 indicated wall-following as their dominant method, and all 10 reported that the layout of the environment was apparent to them only by the end of the test.

We introduce the above problem—of self-organization through exploration—not to provide any particular solutions but to highlight outstanding challenges for the three convergent formulations of sentient behavior considered in this synthesis. The problems of exploring the world and resolving uncertainty—about the exogenous forces and fluctuations encountered—is an inherent part of any self-organized behavior, ranging from how we explore our environment in the visual domain, with saccadic eye movements, through to physical exploration and learning about what would happen if I did that [46]. These are challenging and deep issues that should, at some level, be resolved from a synergetic and free energy principle perspective.

In neurobiology, some progress has been made in terms of minimizing the free energy expected following an action on the world. The functional form of the free energy functional means that, when data are treated as a random variable (from the future), free energy minimization entails a resolution of uncertainty or information gain [47]. This may address the exploratory and epistemic aspect of self-organization. In terms of scaling up this kind of formulation to the level considered by SIRNIA, provisional work using variational free energy has started to address the exchange of an agent with the environment–and indeed other agents. This usually rests upon some form of niche construction and joint minimization of free energy—in the sense that interacting agents (or interactions between a phenotype and her environment) can be seen as a mutual destruction of free energy gradients as agents (and environments) self-organize to learn about each other, e.g., References [48,49,50]. We will not pursue this here but suggest that the formal similarities between the different approaches addressed in this article mean that the synergetic second foundation and free energy minimization should underwrite the scale free aspects of self-organization among conspecifics, communities, and ensembles at any scale.

## 7. Conclusions and Outlook

Homeostasis requires the internal and external regulation of all needed life processes to guarantee the maintenance of order in contrast to thermodynamics with its second law of increasing entropy. An answer to this puzzle was given by Schrödinger: a biological system is an open system whose structure and function is maintained by free energy—or, more precisely, as outlined above, by an influx of free energy.

The requirement of homeostasis entails specific kinds of animal behavior, e.g., acquisition of food. Friston’s FEP may provide a general principle for homeostasis provided the quantity “free energy” is replaced by other quantities, such as, e.g., in brain theory, the firing rate of a group of neurons. 

But is homeostasis the only characteristics of life? In fact, there is another class of life processes, namely, on the ontological level, the development of an individual, and on the phylogenetic level the evolution of species. 

Here, we draw attention to the pronounced transitions in the development. These may be dramatic changes of structure (morphogenesis) and/or of function/behavior. To cope with morphogenesis, there are models on the formation of patterns on furs, on sea shells, segment formation of insects, etc. Pattern formation is also observed in the inanimate world, such as in fluids (think of the Bénard convection instability) and in chemical reactions (e.g., “Belousov–Zhabotinsky”). In physics, chemistry, and biology, the basic explanation of these processes can be traced back to Turing’s [51] model, which was further developed by Prigogine and Nicolis [52] by the formulation of diffusion-reaction equations.

A systematic, unified approach to the solution of these equations has been developed by Synergetics 1st Foundation based on the concept of order parameters and the slaving principle. In this way, *hierarchies* of structures have been dealt with, e.g., in lasers and fluids. When a specific control parameter C (e.g., energy influx) is increased more and more, at specific values of C, a structure is replaced by a new one. The focus of Synergetics 1st Foundation was in the formation of *dynamic* structures: they collapse when the influx of energy and/or matter is stopped. Thus, these systems cannot establish solid structures nor may they process some kind of memory. In physics, solidification, such as crystal growth, growth of dendrites, etc., have been modeled. 

We are confronted with the requirement of a theory that deals with biological phase transitions, such as from a caterpillar to a butterfly, to mention a striking example. Such a theory must consider the interplay between dynamic and solid structures taking into account the fundamental role of information, and show how this process is governed by principles of self-organization/Synergetics.

## Figures and Tables

**Figure 1 entropy-23-00707-f001:**
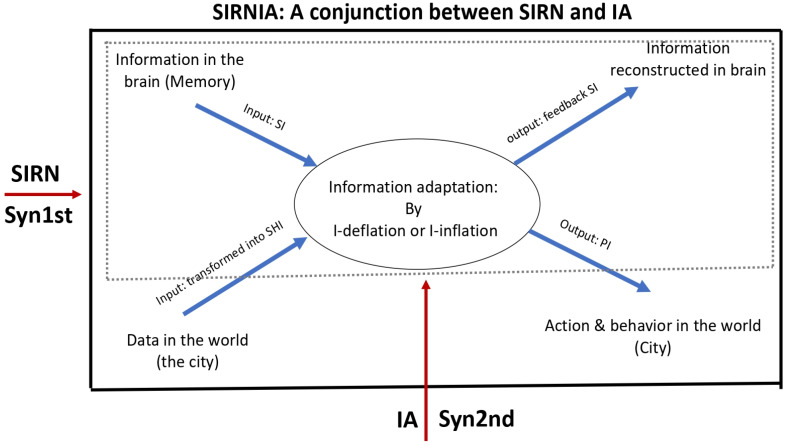
The SIRNIA basic model. For details, see text.

**Figure 2 entropy-23-00707-f002:**
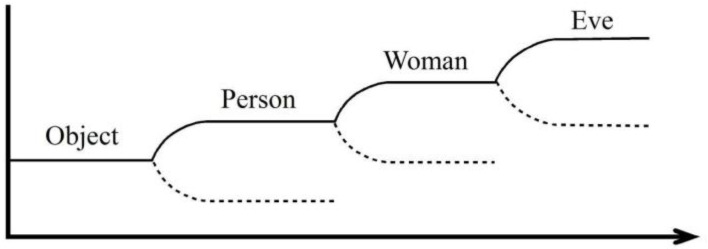
Here is a thought experiment. Imagine a sequence of phase transitions in the approaching lady thought experiment (abscissa = amount of data vs. ordinate = recognized category/pattern). The broken line indicates the other dismissed options. For details, see Reference [1].

**Figure 3 entropy-23-00707-f003:**
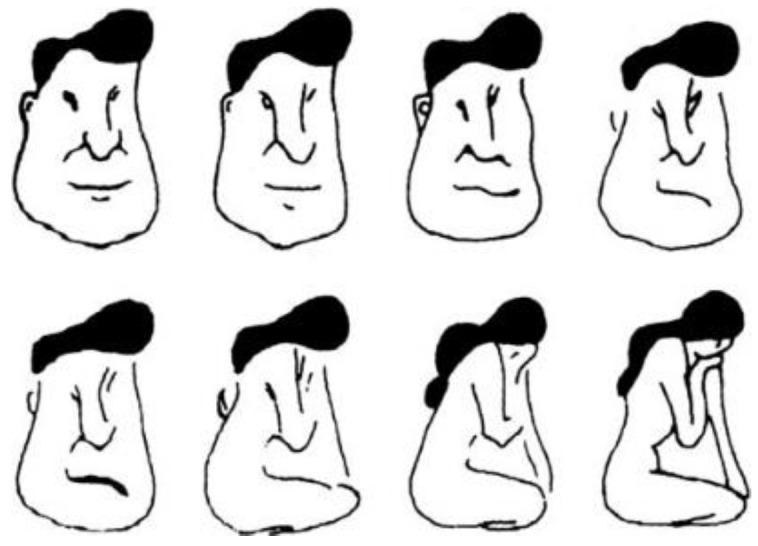
Hysteresis in pattern recognition. When the sequence of figures is visually scanned from the upper left to the lower right, the switch from a face to a girl occurs in the lower row. When scanned in the reverse direction, the switch occurs in the upper row.

**Figure 4 entropy-23-00707-f004:**
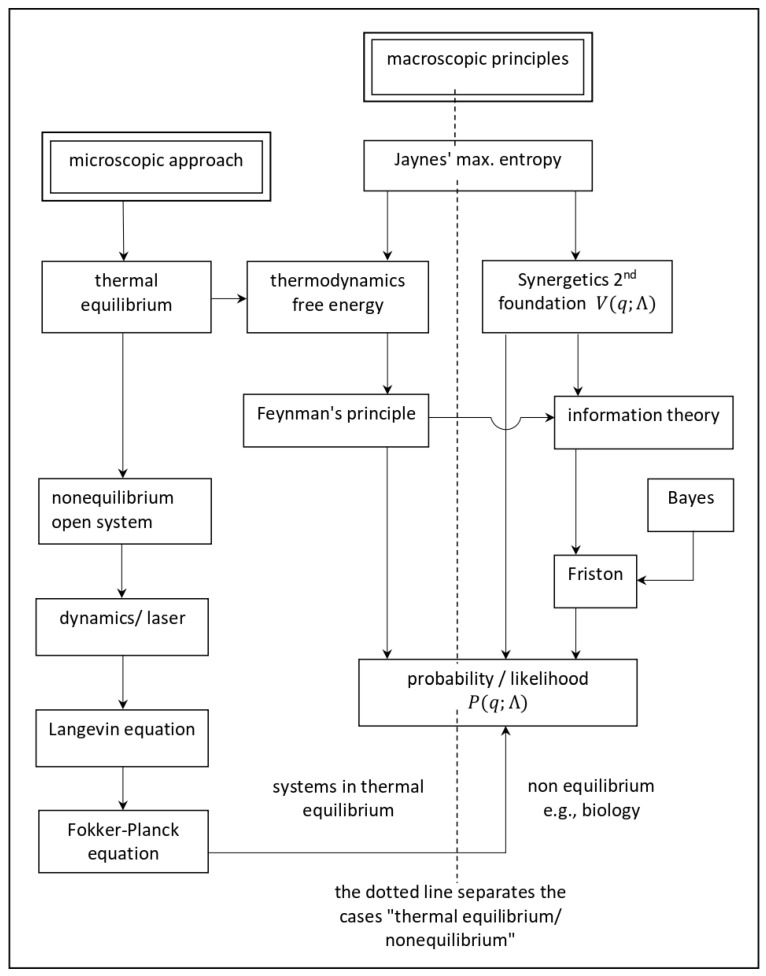
Pathways to (*q*; Λ). This table shows the logical connection between the various approaches. The left column refers to the microscopic theory with its focus on open systems and its mathematical tools, i.e., Langevin and Fokker-Planck equations. The solution of the latter is the probability distribution *P*. The middle part is based on Jaynes’ maximum (information) entropy principle, which contains Feynman’s principle and Synergetics 2nd Foundation as special cases. Both, jointly with Bayesian inference, form a basis of Friston’s FEP. A common denominator of these approaches is the probability *P*, but now derived in a top-down fashion.

**Figure 5 entropy-23-00707-f005:**
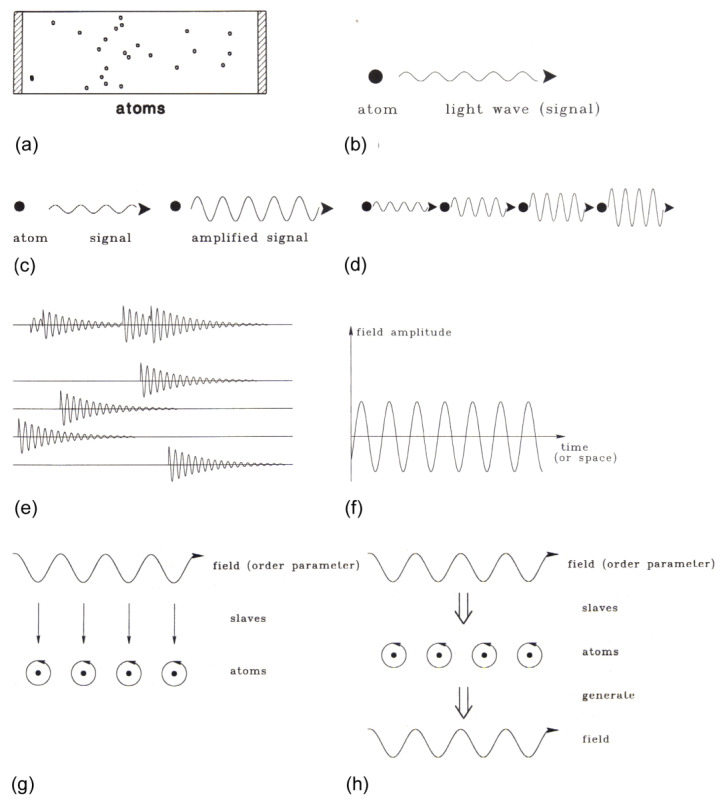
The laser paradigm. (**a**) Typical setup of a gas laser. A glass tube is filled with gas atoms, and two mirrors are mounted at its end faces. The gas atoms are excited by an electric discharge. Through one of the semi-reflecting mirrors, the laser light is emitted. (**b**) An excited atom emits light wave (signal). (**c**) When the light wave hits an excited atom, it may cause the atom to amplify the original light wave. (**d**) A cascade of amplifying processes. (**e**) The incoherent superposition of amplified light waves produces still rather irregular light emission (as in a conventional lamp). When sufficiently many waves are amplified, they strongly compete for further energetic supply. That wave that amplifies fastest wins the competition initiating laser action. (**f**) In the laser, the field amplitude is represented by a sinusoidal wave with practically stable amplitude and only small phase fluctuations. The result: a highly ordered, i.e., coherent, light wave is generated. (**g**) Illustration of the slaving principle. The field acts as an order parameter and prescribes the motion of the electrons in the atoms. Thus, the motion of the electrons is “enslaved” by the field. (**h**) Illustration of circular causality. On the one hand, the field acting as order parameter enslaves the atoms. On the other hand, the atoms by their stimulated emission generate the field [5] (Figure 3.2).

**Figure 6 entropy-23-00707-f006:**
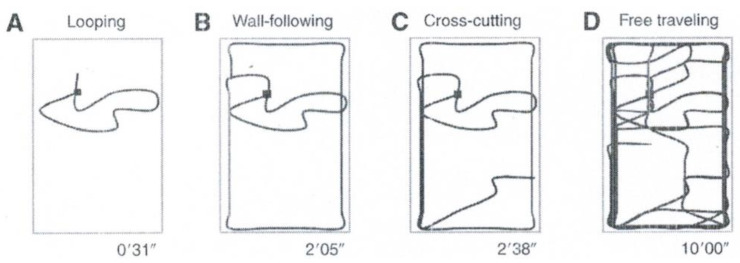
Four cumulative stages in the exploratory behavior of a blindfolded female subject in an unfamiliar environment. Four exploratory behaviors are represented from left to right. The end time of each phase is depicted at the bottom of each plot. The start point of testing is marked by the filled square. As can be seen, this subject started by progressing counterclockwise closing a **loop** (**A**). During this travel, she arrived twice at the room’s wall. Next, she **followed the walls**, encompassing the entire room (**B**). Then, she performed a **cross-cut** (**C**) and, finally, **traveled freely** across the test room (**D**). Source: Reference [42], Figure 1.

## Data Availability

Not relevant.

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
