# Peer review of "Information and Self-Organization II: Steady State and Phase Transition"

_entropy, 2021, doi:10.3390/e23060707_

Round 1
Reviewer 1 Report
I enjoyed reading this compelling synthesis of synergetics, the free energy principle and Shannon information theoretic approaches to self organisation. This is a beautiful and thoughtful synthesis that is written in an accessible way. I think many readers will find it most useful. I have a few suggestions that might improve the presentation of your ideas. Perhaps you could consider the following:
Major points
Page 25, Line 900: it would be nice to link your ideas to the literature on predictive processing in cognitive neuroscience. This is often cast in terms of prediction error minimisation. Your formulation gracefully covers this account of predictive processing and coding. I would recommend the following text after Line 909:
“The above formulation of free energy minimisation is often cast as prediction error minimisation (Rao and Ballard 1999, Clark 2013, Hohwy 2013). This is especially true for generative models with additive Gaussian noise. In this setting, the mechanics we have been describing reduces to predictive coding or Kalman filtering. The focus on the prediction error minimisation – in the cognitive neurosciences (and beyond) – is fully licensed by the general observation that the gradients of free energy can always be expressed as a prediction error of one form or another; namely, the difference between some sampled sensory data and that predicted under a generative model. In short, minimising free energy can be construed as minimising prediction errors. In turn, this casts self organisation as the ability of certain systems to destroy the free energy gradients that created them (Tschacher and Haken 2007). This synergetics perspective was, in fact, one of the inspirations for the free energy principle (Friston, personal communication)."
Page 25, line 924: it might be nice to emphasise the intimate relationship between the FDP and the synergetics second foundation. Perhaps with an additional paragraph like the following:
“When expressed like this, one can see that there is an intimate relationship between the FEP and synergetics: in the sense that minimising free energy is just an expression of the maximum entropy principle under constraints. In the synergetics formulation, these constraints are supplied in the form of Lagrange multipliers that inherit from the physics of the problem at hand. In the free energy principle these constraints are expressed in terms of prior constraints under a Bayesian generative model. In both instances the imperative is to provide an accurate account of – usually sparse – data while maintaining the maximum entropy of that account (i.e., maximising the entropy of an approximate posterior density). One could conceive of the maximum entropy principle as furnishing a unique solution to an inference or measurement problem in the spirit of Occam's principle."
Page 27: section 6.2.2 and section 6.2.3
I found these two sections a bit too brief and difficult to assimilate into the narrative of the previous sections. My recommendation would be to remove these two sections and motivate the nice example of the previous section with something like the following:
“We introduce the above problem – of self organisation through exploration – not to provide any particular solutions but to highlight outstanding challenges for the three convergent formulations of sentient behaviour considered in this synthesis. The problems of exploring the world and resolving uncertainty – about the exogenous forces and fluctuations encountered – is an inherent part of any self-organised behaviour; ranging from how we explore our environment in the visual domain, with saccadic eye movements, through to physical exploration and learning about what would happen if I did that (Schmidhuber 1991). These are challenging and deep issues that should, at some level, be resolved from a synergetic and free energy principle perspective.
In neurobiology, some progress has been made in terms of minimising the free energy expected following an action on the world. The functional form of the free energy functional means that when data are treated as a random variable (from the future), free energy minimisation entails a resolution of uncertainty or information gain (Friston, FitzGerald et al. 2017). This may address the exploratory and epistemic aspect of self organisation. In terms of scaling up this kind of formulation to the level considered by SIRNIA, provisional work using variational free energy has started to address the exchange of an agent with the environment – and indeed other agents. This usually rests upon some form of niche construction and joint minimisation of free energy – in the sense that interacting agents (or interactions between a phenotype and her environment) can be seen as a mutual destruction of free energy gradients as agents (and environments) self organise to learn about each other: e.g.,(Bruineberg, Rietveld et al. 2018, Constant, Ramstead et al. 2018, Kuchling, Friston et al. 2019). We will not pursue this here but suggest that the formal similarities between the different approaches addressed in this article mean that the synergetic second foundation and free energy minimisation should underwrite the scale free aspects of self organisation among conspecifics, communities and ensembles at any scale."
Minor points
Page 3, line 112. I would replace "conveyed" with "furnished"
Page 4, line 147. To make it easy for the reader, spell out CAS (complex adaptive systems).
Page 4, line 154. Change "devise" to "device"
Page 4, line 163. Replace "going on" with "ongoing"
Page 5, line 179. Delete "in task"
Page 5, line 183. change "case" to "cases"
Page 6, line 220. I liked the quotes from Schrodinger about free energy.
Page 6, line 237. I think James et al. [18] should be Ramstead et al. James is his first name.
Page 9, line 317. Please delete "also"
Page 9, line 322. Perhaps change "having said this" with "having established this,"
Page 9, line 329. replace "long enough" with "over a sufficient period of time"
Page 10, line 373. it might be worth adding (possibly as a footnote).
"These fluctuating forces change very quickly and are sometimes referred to as random fluctuations."
Page 10, line 385. I think there may be a notational problem with the superscripts and dashes on the second mu?
Page 11, line 414. Replace “a wanted" with "a requisite"
Page 12, line 436 replace "finally resulting equation "with final result"
Page 12, line 447. It might be worth spelling out that F_tot is the total fluctuation or exogenous forcing term.
Page 12, line 457. Replace "we can do” with "we can implement"
Page 13, line 473. It might be worthwhile with reminding the reader that Q_tot is the amplitude of the total fluctuations. You can then relate this to the energy flux due to noise on line 480.
Page 17, line 644 and 646. Could you replace “II” in equations 23 and 24 with double bars.
Page 19, line 688. Perhaps replace "outsprings" with "offsprings" or, possibly, "sequelae".
Page 20, line 734 it might be useful to define Dkl(lambda) explicitly. Noting that the KL divergence can be expressed as a function of the sufficient statistics (lambda) of two probability densities.
Page 22, line 796. You can also add: "offered by the use of Gaussians – often known as the Laplace assumption in Bayesian statistics (Friston, Mattout et al. 2007)"
Page 22, line 815. Please change "feasibly" to "feasible"
Page 24, line 864. when you mention equation numbers these appear to refer to the previous section. It might be useful to make this explicit?
Page 25 line 888. Please check the spelling of “executed”.
I hope these comments help should any revision be required.
Bruineberg, J., E. Rietveld, T. Parr, L. van Maanen and K. J. Friston (2018). "Free-energy minimization in joint agent-environment systems: A niche construction perspective." Journal of Theoretical Biology 455: 161-178.
Clark, A. (2013). "Whatever next? Predictive brains, situated agents, and the future of cognitive science." Behav Brain Sci. 36(3): 181-204.
Constant, A., M. J. D. Ramstead, S. P. L. Veissiere, J. O. Campbell and K. J. Friston (2018). "A variational approach to niche construction." J R Soc Interface 15(141).
Friston, K., T. FitzGerald, F. Rigoli, P. Schwartenbeck and G. Pezzulo (2017). "Active Inference: A Process Theory." Neural Comput 29(1): 1-49.
Friston, K., J. Mattout, N. Trujillo-Barreto, J. Ashburner and W. Penny (2007). "Variational free energy and the Laplace approximation." NeuroImage 34(1): 220-234.
Hohwy, J. (2013). The Predictive Mind. Oxford, Oxford University Press.
Kuchling, F., K. Friston, G. Georgiev and M. Levin (2019). "Morphogenesis as Bayesian inference: A variational approach to pattern formation and control in complex biological systems." Phys Life Rev.
Rao, R. P. and D. H. Ballard (1999). "Predictive coding in the visual cortex: a functional interpretation of some extra-classical receptive-field effects." Nat Neurosci. 2(1): 79-87.
Schmidhuber, J. (1991). "Curious model-building control systems." In Proc. International Joint Conference on Neural Networks, Singapore. IEEE 2: 1458–1463.
Tschacher, W. and H. Haken (2007). "Intentionality in non-equilibrium systems? The functional aspects of self-organised pattern formation." New Ideas in Psychology 25: 1-15.
Author Response
Dear reviewer
We are most grateful to you for your substantial suggestions improving our manuscript, the detailed lists of errors corrections and the additional references. We have taken care of all of that (in Red).
Reviewer 2 Report
The paper is written in a clear and consistent form. Bibliography is full and very helpful. The paper will be useful for a broad audience of students, post-docs and researchers. The research carried out by the authors is valuable and worth publishing in "Entropy".
Minor corrections to be made:
- page 8, line 305: “maximum(information)”, separate the words.
- page 10, line 364: the dot symbol is above the line.
- page 11, line 409: correct the position of: I II and III.
- page 12, line 467: Missing parentheses “)” in Eq. (14).
- page 14, line 513: zylinder => cylinder.
- page 14, line 544: “N1dashes”, separate.
- page 24, line 858: “P(psi|r) .” eliminate space.
- page 25, line 888: correct the word "exscuted ".
- page 29, line 1074: eliminate the dots.
- Adapt references to the style of the journal.
Author Response
Dear reviewer
Many thanks for reviewing our paper and your error corrections, we have taken care of (in Red).